# Bicarbonate-Independent Sodium Conductance of Na/HCO$_3$ Cotransporter NBCn1 Decreases NMDA Receptor Function

**Inyeong Choi \*** , **Hansoo Yang, Eunjin Kim and Soojung Lee**

Department of Cell Biology, Emory University School of Medicine, Atlanta, GA 30322, USA;
hansooyang@outlook.com (H.Y.); eunjin0825@hotmail.com (E.K.); soojung.lee@gatech.edu (S.L.)
\* Correspondence: ichoi@emory.edu; Tel.: +1-404-712-2092

**Abstract:** The sodium bicarbonate cotransporter NBCn1 is an electroneutral transporter with a channel activity that conducts Na$^+$ in a HCO$_3^-$-independent manner. This channel activity was suggested to functionally affect other membrane proteins which permeate Na$^+$ influx. We previously reported that NBCn1 is associated with the NMDA receptors (NMDARs) at the molecular and physiological levels. In this study, we examined whether NBCn1 channel activity affects NMDAR currents and whether this effect involves the interaction between the two proteins. NBCn1 and the NMDAR subunits GluN1A/GluN2A were expressed in *Xenopus* oocytes, and glutamate currents produced by the receptors were measured using two-electrode voltage clamp. In the absence of CO$_2$/HCO$_3^-$, NBCn1 channel activity decreased glutamate currents mediated by GluN1A/GluN2A. NBCn1 also decreased the slope of the current–voltage relationships for the glutamate current. Similar effects on the glutamate current were observed with and without PSD95, which can cluster NBCn1 and NMDARs. The channel activity was also observed in the presence of CO$_2$/HCO$_3^-$. We conclude that NBCn1 channel activity decreases NMDAR function. Given that NBCn1 knockout mice develop a downregulation of NMDARs, our results are unexpected and suggest that NBCn1 has dual effects on NMDARs. It stabilizes NMDAR expression but decreases receptor function by its Na$^+$ channel activity. The dual effects may play an important role in fine-tuning the regulation of NMDARs in the brain.

**Keywords:** sodium bicarbonate transporter; NBCn1; channel activity; NMDA receptors; intracellular sodium

## 1. Introduction

NBCn1 (SLC4A7) moves Na$^+$ and HCO$_3^-$ across cell membranes and regulates cellular pH and transepithelial HCO$_3^-$ transport in many cells [1–3]. The movement occurs with the stoichiometry of 1 Na$^+$ to 1 HCO$_3^-$, such that both ions enter cells without altering cell membrane potential [4]. In addition to the cotransport, NBCn1 also has a channel activity that produces a HCO$_3^-$-independent Na$^+$ conductance [4,5]. This conductance increases intracellular Na$^+$ levels ([Na$^+$]$_i$) and causes the resting membrane potential to shift positively. The channel activity is not uniquely reported in NBCn1, as channel activities with distinct ion selectivity are also observed in other members of SLC4A [6–8] and SLC26A bicarbonate transporters [9,10]. Due to its Na$^+$ conductance, NBCn1 channel activity is expected to alter a Na$^+$ influx and [Na$^+$]$_i$, which is critical for neuronal function in the nervous system. Nonetheless, the exact functional role of this activity is presently unclear.

In most neurons, NBCn1 is mainly localized to postsynaptic membranes, where it interacts with a variety of proteins to form a macromolecular complex [11–13]. Among the proteins capable of interacting with NBCn1 is the postsynaptic scaffolding protein PSD95 that recruits many postsynaptic proteins including N-methyl-D-aspartic acid receptors (NMDARs) [13]. Animal studies show that NBCn1 and NMDARs appear to be tightly associated with each other, such that a change in NBCn1 levels leads to a change in NMDAR levels [14]. NBCn1 knockout mice develop a downregulation of the NMDAR subunit

GluN1 and PSD95 and are resistant to $Mg^{2+}$/NMDA-mediated seizure and excitotoxicity [14]. The downregulation of NMDARs in these mice may also be responsible for animals' increased alcohol consumption [15], given the importance of glutamatergic transmission for the reward pathway of the brain [16,17]. The downregulation of NMDARs in NBCn1 knockout mice is also consistent with the report [18] that NBCn1 upregulation, induced by acidification, correlates with NMDA-mediated neuronal damage in rat hippocampal primary cultures. NBCn1 is an acid-base regulator, and NMDARs are sensitive to pH; thus, from the perspective of pH physiology, the positive correlation between the two proteins is an advantage for proper regulation of the receptors. On the other hand, it is unclear whether NBCn1 channel activity is also involved.

In this study, we examined the effects of NBCn1 channel activity on NMDAR currents and the contribution of PSD95 to those effects. We expressed GluN1A/GluN2A (hereafter, GluN1·N2) and NBCn1, as well as PSD95, in *Xenopus* oocytes and tested whether currents evoked by glutamate ($I_{Glu}$) were affected by NBCn1 and PSD95. Oocytes were chosen for the expression system as multiple proteins were simultaneously expressed in single cells, which allowed our comparison analysis to be more reliable than other expression systems. The results show that NBCn1 channel activity decreased $I_{Glu}$ mediated by GluN1·N2, and this decrease can serve to compensate an $I_{Glu}$ increase by PSD95. The decreasing effects were observed at comparable levels with and without PSD95. Given the manifestation of NMDAR excitotoxicity and abnormal activity in neurodegenerative diseases, such as Alzheimer's disease [19], we propose that stimulating NBCn1 channel activity in neurons might be a valuable approach to reducing excessive glutamate transmission under pathological conditions.

## 2. Results

### 2.1. NBCn1 Channel Activity Increases the Baseline Conductance and Positively Shifts the Resting Membrane Potentials of Xenopus Oocytes

Figure 1A shows current–voltage (*I–V*) relationships for the baseline conductance in oocytes expressing NBCn1 or water-injected controls, measured at holding potentials from −120 to +60 mV in 20 mV steps. Experiments were performed in $CO_2$/$HCO_3^-$-free ND96 solution containing 96 mM NaCl, as the channel activity is independent of bath $HCO_3^-$. Compared to the control, oocytes expressing NBCn1 produced a higher slope of the *I–V* plot (*n* = 5–7/group), a hallmark for its channel activity [4,5]. The inward currents at negative voltages are due to the $Na^+$, whereas the outward currents at positive voltages are due to other ions caused by the $Na^+$ influx [4,5]. Figure 1B shows the $Na^+$ component in the *I–V* plot, obtained from the difference before and after $Na^+$ removal from the bath. Inward currents progressively increased at more negative voltages and the reversal potential approached to +60 mV, close to the reversal potential for a $Na^+$ channel. In addition, NBCn1 channel activity caused the resting membrane potentials to be more positive (*p* < 0.01; Figure 1C).

### 2.2. NBCn1 Channel Activity Decreases $I_{Glu}$ Produced by GluN1·N2

To examine whether the above activity affects NMDAR function, we expressed GluN1·N2 with or without NBCn1 in oocytes and compared $I_{Glu}$ between the two groups. Figure 2A shows a representative $I_{Glu}$ evoked by 100 μM glutamate (plus 10 μM glycine, no $Mg^{2+}$) in an oocyte expressing GluN1·N2. A rapid inward current followed by a steady-state current was recorded, characteristic for GluN1·N2. Interestingly, $I_{Glu}$ was decreased in the coexpression of NBCn1 (Figure 2B–D). The decrease was small at 1 ng of NBCn1 cRNA for injection, but substantial at 5 and 10 ng for injection. The membrane potentials were progressively more positive at higher amounts of NBCn1 cRNAs, consistent with increasing channel activities (data not shown). A control had no $I_{Glu}$ (Figure 2E). Comparison of the mean $I_{Glu}$ obtained from multiple oocytes confirmed this inhibitory effect of NBCn1 on GluN1·N2 (*p* < 0.05, one-way ANOVA, *n* = 4–5/group; Figure 2F).

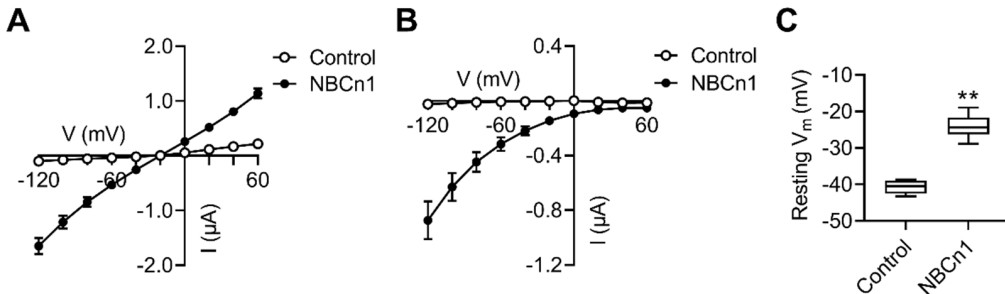

**Figure 1.** NBCn1 channel activity. (**A**) *I–V* relationships in oocytes expressing NBCn1 or water-injected control oocytes. Oocytes were subjected to step-voltage commands from –120 to +60 mV (20 mV steps) in $CO_2/HCO_3^-$-free ND96 solution (*n* = 5–7/group). The increased slope of the *I–V* plot is a hallmark for NBCn1 channel activity. (**B**) $I_{NBCn1}$–*V* relationship of NBCn1 channel activity. The plot was obtained from the difference between the *I–V* plots in the presence and absence of bath $Na^+$. (**C**) Resting membrane potentials ($V_m$). $V_m$ was measured 48–72 h after cRNA injection (*n* = 5 controls and 16 NBCn1). ** $p < 0.01$, Student *t*-test.

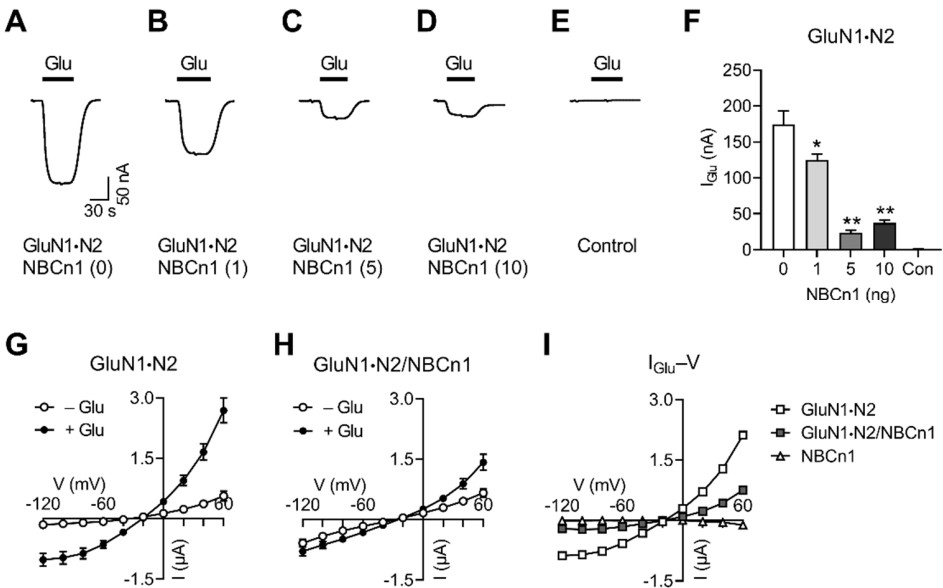

**Figure 2.** NBCn1 channel activity decreases $I_{Glu}$ produced by GluN1·N2. (**A**) Representative $I_{Glu}$ produced by GluN1·N2. $I_{Glu}$ was measured with an application of 100 μM glutamate (10 μM glycine, no $Mg^{2+}$). Recording was performed in $CO_2/HCO_3^-$-free ND96 solution containing 96 mM NaCl (holding potential of –40 mV). (**B–D**) Representative $I_{Glu}$ produced by GluN1·N2/NBCn1. A fixed amount of GluN1·N2 was coexpressed with 1, 5, or 10 ng of NBCn1. (**E**) Control. (**F**) Mean $I_{Glu}$. Data were obtained from 4–5 oocytes/group. * $p < 0.05$, ** $p < 0.01$ compared to GluN1·N2 alone, one-way ANOVA with Dunnett post-test. (**G**) and (**H**) *I–V* relationships of glutamate-evoked responses by GluN1·N2 (*n* = 6; (**G**)) and GluN1·N2/NBCn1 (*n* = 5; (**H**)). (**I**) $I_{Glu}$–*V* relationships. $I_{Glu}$ is the mean difference before and after glutamate application in (**G**) and (**H**). NBCn1 alone served as a control (*n* = 4).

The decreasing effect of NBCn1 on $I_{Glu}$ was further evaluated by *I–V* relationships. Figure 2G shows the *I–V* plots for the responses evoked by glutamate in oocytes expressing GluN1·N2 (*n* = 6). The response yielded a plot that is characteristic for glutamate *I–V*; that is, while currents had a small increase in amplitude as the membrane is hyperpolarized, there were markedly increased currents as the membrane is depolarized, producing a large increase in the slope conductance (measured at zero-current voltage). Figure 2H shows the *I–V* plots in oocytes expressing GluN1·N2/NBCn1 (*n* = 5). As expected, NBCn1 channel activity increased inward currents at negative potentials in the baseline *I–V* plot.

Glutamate application produced $I_{Glu}$; however, the magnitude was small, compared to that for GluN1·N2 alone in Figure 2G. Thus, the difference between the two *I–V* plots before and after glutamate application (i.e., $I_{Glu}$–*V* relationships) was smaller when NBCn1 was coexpressed, as shown in Figure 2I. Comparison of the $I_{Glu}$–*V* plots between GluN1·N2 and GluN1·N2/NBCn1 resulted in a significant reduction in the slope by NBCn1 ($F_{9,90}$ = 26.34, $p < 0.01$, two-way ANOVA repeated measures). The control with NBCn1 alone showed negligible $I_{Glu}$ ($n = 4$). Together, these results demonstrate that NBCn1 channel activity decreases $I_{Glu}$ mediated by GluN1·N2.

### 2.3. NBCn1 Channel Activity Compensates an $I_{Glu}$ Increase Induced by PSD95

NBCn1 can cluster with NMDARs via PSD95 [11,12]. As PSD95 can affect both NMDARs [20] and NBCn1 [13], its contribution to the abovementioned NBCn1-mediated effects on $I_{Glu}$ needs to be addressed. For this task, we performed two sets of experiments. In the first set of experiments, we examined the effect of PSD95 on $I_{Glu}$ produced by GluN1·N2. As shown in Figure 3, PSD95 increased $I_{Glu}$ and induced a steeper slope of the $I_{Glu}$–*V* relationships. The two $I_{Glu}$–*V* plots in Figure 3C crossed at the reversal potential, indicating that the current increase was mediated by GluN1·N2, not by any other current source. The result is consistent with the report [21] that PSD95 enhances NMDAR function by increasing the rate of channel insertion to membrane.

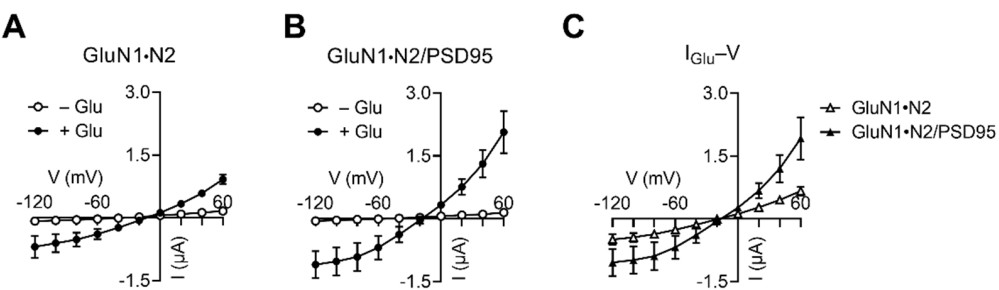

**Figure 3.** PSD95 increases $I_{Glu}$ produced by GluN1·N2. (**A**) *I–V* relationships of glutamate-evoked responses by GluN1·N2 ($n = 3$). (**B**) *I–V* relationships of glutamate-evoked responses by GluN1·N2/PSD95 ($n = 5$). (**C**) $I_{Glu}$–*V* relationships. $I_{Glu}$ is the mean difference before and after glutamate application in (**A**) and (**B**).

In the second set of experiments, we compared $I_{Glu}$ produced by GluN1·N2/NBCn1 vs. GluN1·N2/NBCn1/PSD95. Interestingly, both groups produced similar responses to glutamate application, and their $I_{Glu}$–*V* plots were similar (Figure 4). The reversal potentials were slightly separated, the reason of which is unclear, although we think the separation might be related to the difference between non-clustered vs. clustered protein complexes. Regardless, the two plots were nearly superimposed, indicating no PSD95 effect on $I_{Glu}$ when NBCn1 was present. Together, the results from the two sets of experiments demonstrate that NBCn1 channel activity counteracts the effect of PSD95 on NMDAR function.

### 2.4. $I_{Glu}$ Decrease by NBCn1 Channel Activity Also Occurs in the Presence of $CO_2/HCO_3^-$

The above experiments were performed in the absence of $CO_2/HCO_3^-$. To examine whether the decreasing effect of the channel activity on $I_{Glu}$ also occurs in the presence of $CO_2/HCO_3^-$, we measured $I_{Glu}$ before and 10 min after applying a solution equilibrated with 10% $CO_2$, 50 mM $HCO_3^-$ at constant pH 7.4 and compared them. Except when glutamate was applied, $Na^+$ in $CO_2/HCO_3^-$ solution was replaced with N-methyl glucamine to minimize the cotransport activity, which changes $pH_i$ and might complicate the results. Figure 5A shows representative $I_{Glu}$ traces produced by GluN1·N2/PSD95. The $I_{Glu}$ with similar amplitudes were observed before and after applying $CO_2/HCO_3^-$. Figure 5B shows $I_{Glu}$ traces from a parallel experiment with GluN1·N2/PSD95/NBCn1. The $I_{Glu}$ was smaller than that for GluN1·N2/PSD95, which is expected from the inhibitory effect of NBCn1.

Nonetheless, $I_{Glu}$ were similar before and after applying $CO_2/HCO_3^-$, comparable to those for GluN1·N2/PSD95. Consistently, the mean $I_{Glu}$ from multiple oocytes resulted in no significant difference before and after $CO_2/HCO_3^-$ application within groups ($p > 0.05$, paired two tailed Student t-test; $n = 4$–6/group; Figure 5C,D). NBCn1 decreased $I_{Glu}$ by $43 \pm 7\%$ and $49 \pm 9\%$ in the absence and presence of $CO_2/HCO_3^-$, respectively (Figure 5E). Conclusively, NBCn1 channel activity decreased $I_{Glu}$ regardless of bath $CO_2/HCO_3^-$.

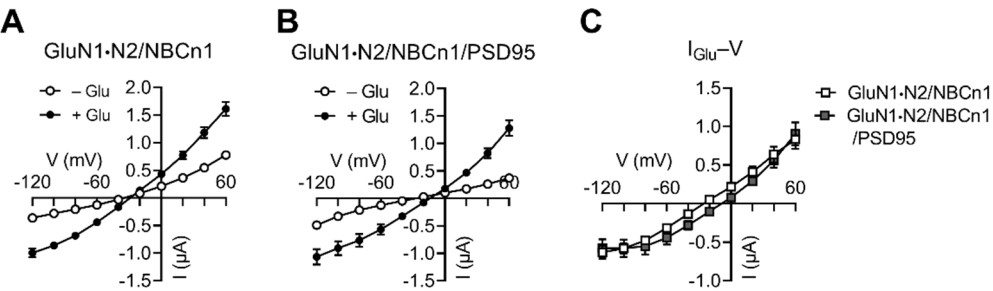

**Figure 4.** PSD95 has negligible effect on $I_{Glu}$ in the presence of NBCn1. (**A**) *I–V* relationships of glutamate-evoked responses by GluN1·N2/NBCn1 ($n = 11$). (**B**) *I–V* relationships of glutamate-evoked responses by GluN1·N2/PSD95/NBCn1 ($n = 7$). (**C**) $I_{Glu}$–*V* relationships. $I_{Glu}$ is the mean difference before and after glutamate application in (**A**,**B**).

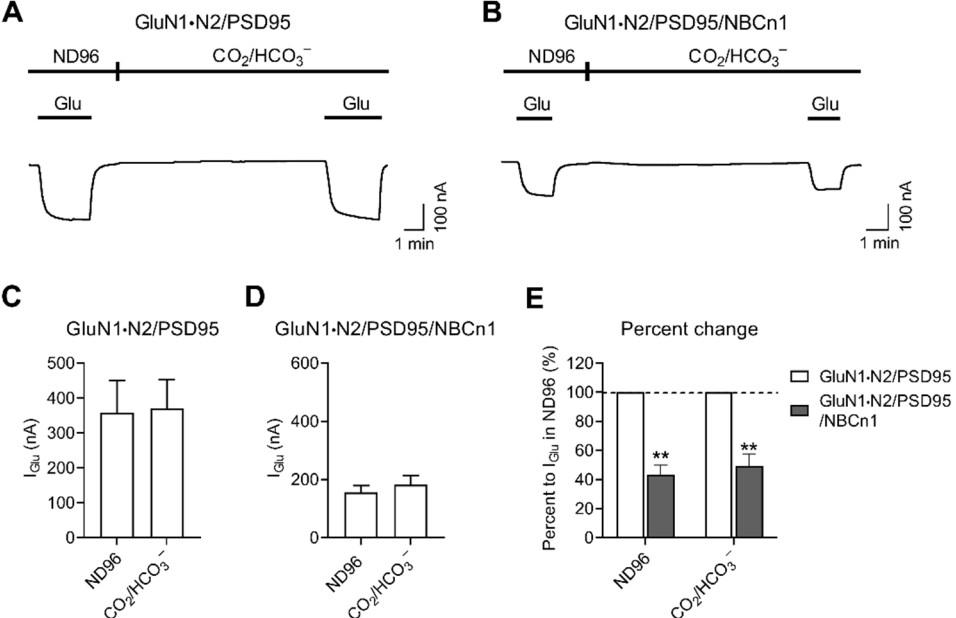

**Figure 5.** The $I_{Glu}$ decrease by NBCn1 channel activity is independent of $CO_2/HCO_3^-$. (**A**,**B**) Representative $I_{Glu}$ produced by GluN1·N2/PSD95 (**A**) and GluN1·N2/PSD95/NBCn1 (**B**). $I_{Glu}$ was measured in ND96 solution and 10 min after applying a solution equilibrated with 10% $CO_2$, 50 mM $HCO_3^-$ at constant pH 7.4. Except when glutamate was applied, $Na^+$-free $CO_2/HCO_3^-$ solution was applied to minimize $pH_i$ change by cotransport activity. (**C**) Mean $I_{Glu}$ produced by GluN1·N2/PSD95 ($n = 4$; (**C**)) and GluN1·N2/PSD95/NBCn1 ($n = 6$; (**D**)). (**E**) Effects of NBCn1 on $I_{Glu}$ in the absence and presence of $CO_2/HCO_3^-$. Data were presented as percent change relative to $I_{Glu}$ by GluN1·N2/PSD95. ** $p < 0.01$.

## 3. Discussion

In this study, we found that NBCn1 channel activity decreases $I_{Glu}$ produced by GluN1·N2 function. The channel activity induces 45–60% of the expected membrane potential change per $Na^+$ decade based on the Nernst equation [4,5]. External $K^+$, $Mg^{2+}$, $Ca^{2+}$, or internal $Cl^-$ do not affect the activity [5]. This leads to the possibility that roughly

half of the expected Nernst value reflects two binding sites for Na$^+$, rather than one Na$^+$ and an additional ion contributing the remaining half of the conductance. A hint comes from the ion transport through the similar transporter, NBCe1, that carries CO$_3^{2-}$ [22,23]. In particular, a recent report on the molecular dynamics simulations of NBCe1 [24] proposes a potential binding site for CO$_3^{2-}$. Given the structural similarity between NBCe1 and NBCn1 (>65%) and a charge conservation in the binding site, it is possible that NBCn1 transports 2 Na$^+$ and 1 CO$_3^{2-}$, and the channel activity is related to a leak at the Na$^+$ binding site. We have a preliminary observation that a mutation in the site associated with the Na$^+$ binding abolishes the channel activity (unpublished data). Furthermore, the channel activity is enhanced by the stilbene derivate DIDS [4,5], which inhibits all sodium bicarbonate transporters except NBCn1 [1,25]. As DIDS occludes the entrance of the ion passageway in NBCe1 [24], the channel activity in NBCn1 may occur through the same ion passageway without being coupled to the cotransport activity. It will be interesting to investigate the molecular nature of the channel activity for future study.

NBCn1 channel activity increases Na$^+$ influx and raises [Na$^+$]$_i$. It is thus conceivable that the channel activity reduces an electrochemical gradient of Na$^+$ with the consequence of a decrease in the driving force for Na$^+$ influx via NMDARs. On the other hand, our results are inconsistent with others [26,27] that report that an increase in [Na$^+$]$_i$ enhances NMDAR currents and single channel activity in cultured spinal and hippocampal neurons. These changes are mediated by increasing Ca$^{2+}$ influx through NMDARs and overcoming the Ca$^{2+}$-dependent inactivation of NMDAR gating. Similarly, an increase in [Na$^+$]$_i$ by a voltage-gated sodium channel modifier increases Ca$^{2+}$ influx through NMDARs and engages activity-dependent Ca$^{2+}$ signaling mechanisms that lead to structural plasticity in cerebrocortical neurons [28]. It is unclear why there is a disparity between our study and others. One possible reason for this disparity is protein tyrosine kinase Src, which is associated with NMDAR regulation [29]. Src is involved in the receptor sensitivity to [Na$^+$]$_i$ [26,27], and its expression levels are high in most neurons [30]. Sato et al. [31] reported that the Src expression level is very low in *Xenopus* oocytes, below the detection limit. If the disparity between our study and others is due to Src, then our study supports the idea that intracellular Na$^+$ does not directly affect NMDARs, but instead serves as a signaling ion. Another possible reason is the magnitude of the [Na$^+$]$_i$ change by NBCn1. With an injection of >25 ng of NBCn1 cRNA per oocyte, the channel activity can raise [Na$^+$]$_i$ by ≈40 mM [4], sufficient to trigger NMDARs activation [26,27]. However, we could not use this amount of NBCn1 for coexpression with GluN1·N2 and PSD95, because *Xenopus* oocytes have a limited capacity to translate membrane-bound mRNAs [32].

Regardless of the reason for the disparity, our results lead to a question of whether the decreasing effect of NBCn1 channel activity on NMDARs can be recapitulated in neurons. We do not rule out the possibility that NBCn1 channel activity conversely enhances NMDAR function in neurons. Neurons express a variety of Na$^+$-permeable channels, such as voltage-gated Na$^+$ channels [33,34] and Na$^+$ leak channels [35], and changing [Na$^+$]$_i$ beyond the resting level may occur by a wide range of physiological and pathophysiological stimuli. We envision that NBCn1 channel activity will also play a role in regulating [Na$^+$]$_i$ in the CNS.

NBCn1 channel activity decreases NMDAR function in the absence of PSD95 (Figure 2), and there is no apparent difference in $I_{Glu}$ between GluN1·N2/NBCn1 vs. GluN1·N2/NBCn1/PSD95 (Figure 4). Thus, NBCn1 does not need to cluster with NMDARs to decrease receptor function. On the other hand, the clustering is critical for receptor expression and activity, because NBCn1 KO mice develop a significant downregulation of GluN1 and PSD95 in neurons [14]. Neurons from these mice display lower apoptotic cell death associated with excitotoxicity, and their membranes are less excitable. The downregulation of GluN1 and PSD95, thus, demonstrates that NBCn1 is important for NMDAR expression during mouse development. Based on these results, we propose that NBCn1 has dual effects on NMDAR. On the one hand, it stabilizes NMDAR expression by constituting a large protein complex with the receptors; on the other hand, it decreases NMDAR function by its Na$^+$ channel activity.

By producing such dual effects, the transporter can fine-tune the regulation of NMDARs in the brain.

The decreased effect on the receptors by NBCn1 channel activity also occurs in the presence of $CO_2/HCO_3^-$ (Figure 5). The decrease was evident between the absence vs. presence of NBCn1, while there was no difference before and after $CO_2/HCO_3^-$ application. $CO_2/HCO_3^-$ causes intracellular acidification as $CO_2$ enters cells, associates with $H_2O$, and produces $H^+$ and $HCO_3^-$. The $pH_i$ drops to 6.9–7.1 in control oocytes, while it is higher in NBCn1-expressing oocytes [12,36,37]. We inhibited NBCn1 cotransport activity by removing bath $Na^+$ and adding it only when glutamate was applied. There was no difference in $I_{Glu}$ before and after $CO_2/HCO_3^-$ application, thus indicating that a decrease in $pH_i$ by 0.5 unit has negligible effect on GluN1·N2. This further indicates that NMDARs are less sensitive to $pH_i$, in contrast to its steep dependence on extracellular pH with the half maximum inhibition of pH 7.3 [38,39]. The pH-sensitive residues are mainly located in the GluN2 N-terminal domain, determined by mutagenesis study [40] and pKa prediction of titratable residues based on the high-pH model of the GluN1-GluN2A structure [41].

What percentage of the $Na^+$ current relative to cotransport would be mediated by the channel activity? This can be estimated by comparing $Na^+$ ion flux ($J_{Na}$) and $I_{NBC}$ under a voltage clamp condition. $J_{Na}$ is calculated from $(dpH/dt)V\beta_t$, where V is an oocyte volume (1 μL) and $\beta_t$ is total buffering power (i.e., intrinsic buffering power ($\beta_i$) + 2.3 × $[HCO_3^-]_i$). We did not determine $\beta_i$, but the reported value was 18.2 mM at $7.05 < pH_i < 7.25$ [42]. In our pH measurement, dpH/dt was $12.6 × 10^{-5}$ pH/sec, and $[HCO_3^-]_i$ was 7.05 in 5% $CO_2/25$ mM $HCO_3^-$ solution. This gives $J_{Na}$ of $426.284 × 10^{-14}$ mole/sec. At $-60$ mV, NBCn1 channel activity produces 316.3 nA, which equals to $327.83 × 10^{-14}$ mole/sec. Thus, the ratio of $Na^+$ current/cotransport is 0.769; i.e., 76.9% when oocyte membrane is at $-60$ mV. The channel activity produces a $Na^+$ conductance and is markedly reduced as membranes are depolarized.

In summary, our study shows the importance of NBCn1 channel activity for NMDAR regulation. NBCn1 regulates the receptors not only by altering the Na electrochemical gradient, but also by compensating PSD95-mediated effects on receptor function. NBCn1 channel activity provides another mechanism to modulate the receptors, in addition to its cotransport activity. We envision that stimulating NBCn1 channel activity in neurons might be a valuable approach to reducing excitotoxicity and abnormal NMDAR activity in neurodegenerative diseases such as Alzheimer's disease [19]. In this context, it is interesting to note that DIDS or SITS can provide moderate protection against NMDA toxicity and decrease NMDAR-mediated increase in intracellular $Ca^{2+}$ in cultured rat cortical neurons [43]. Overall, our study provides new insights into the functional properties of NBCn1 and offers a basis for future studies on NBCn1-mediated regulation of glutamate transmission in the central nervous system.

## 4. Materials and Methods

### 4.1. Protein Expression in Xenopus Oocytes

Protein expression in *Xenopus* oocytes was performed as described previously [13]. Briefly, *Xenopus laevis* oocytes at stages V–VI were purchased from Ecocyte Bioscience (Austin, TX, USA). NBCn1-E, PSD95, NR1A, and NR2A were transcribed using the mMessage/mMachine transcription kit (Life Technologies, Grand Island, NY, USA). The amount of injected cRNA (in 46 nL) was 14 ng for NBCn1-E, 7 ng for PSD95, 7 ng for NR1A, and 14 ng for NR2A. Thus, the ratio of NR1A to NR2A was 1 to 2. The injection of 46 nL reduced the oocyte osmolality from 300 mOsM to 287 mOsM, which is negligible. Equal amounts of RNAs were used when multiple samples were compared. Controls were injected with sterile water. Oocytes were maintained at 18 °C for 3 days.

### 4.2. Two-Electrode Voltage Clamp

An oocyte was placed in the recording chamber containing ND96 solution (mM; 96 NaCl, 2 KCl, 1.8 $CaCl_2$, 1 $MgCl_2$, 10 HEPES, and pH 7.4) and 1 mM $BaCl_2$ and impaled

with two borosilicate glass electrodes filled with 3 M KCl (tip resistance: 0.5–2 MΩ). After the resting potential was stable, the oocyte was clamped at –60 mV using the voltage-clamp amplifier OC-725C (Warner Instrument, Hamden, CT, USA). For recording $I_{NBCn1}$, current-voltage ($I–V$) relationships were obtained by voltage commands from $-120$ mV to +60 mV (100 msec, in 20 mV increments). Na$^+$-free solution was made by N-methyl-D-glucamine. For recording $I_{Glu}$, the holding potential of –60 mV was used. $I_{Glu}$ was monitored by applying 100 μM glutamate (with 10 μM glycine, but no Mg$^{2+}$). $I–V$ relationships were also obtained by voltage commands in the absence or presence of glutamate application. For recording $I_{Glu}$ in the presence of $CO_2/HCO_3^-$, oocytes were superfused with a solution buffered with 10% $CO_2$, 50 mM $HCO_3^-$, and pH 7.4. Signals were collected using a Digidata 1322 interface (Molecular Devices; Sunnyvale, CA, USA) and analyzed using pClamp 10 (Molecular Devices). Signals were filtered using a Bessel lowpass filter with a cutoff frequency of 0.1 Hz. All experiments were performed at room temperature.

*4.3. Statistical Analysis*

Data were reported as mean ± standard error. The level of significance was determined using (i) unpaired, two-tailed Student t-test for comparison of $I_{Glu}$ between GluN1·N2 vs. GluN1·N2/NBCn1 and comparison of $I_{NBC}$ between control vs. NBCn1; (ii) paired, two-tailed Student t-test for comparison of $I_{Glu}$ before and after $CO_2/HCO_3^-$ application within the same group; (iii) one-way ANOVA for comparison of $I_{Glu}$ among GluN1·N2/NBCn1 with different injection amounts of NBCn1; and (iv) two-way ANOVA for comparison of $I–V$ relationships. The *p* value of less than 0.05 was considered significant. Data were analyzed using Prism 9 (GraphPad; La Jolla, CA, USA) and Microsoft Office Excel add-in Analysis ToolPak (Redmond, WA, USA).

**Author Contributions:** Conceptualization, I.C.; Methodology, H.Y., E.K. and S.L.; Formal analysis, I.C.; Investigation, H.Y., E.K. and S.L.; Writing—original draft preparation, I.C. All authors have read and agreed to the published version of the manuscript.

**Funding:** This research was funded by NIH AA028606 and Emory URC to I.C.

**Data Availability Statement:** The data presented in this study are available upon request to I.C.

**Acknowledgments:** We thank Ana Rivadeneira for editing. The current address for S.L. is the Department of Biomedical Engineering, Georgia Institute of Technology, Atlanta, GA, USA.

**Conflicts of Interest:** The authors declare no conflict of interest.

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
