# Peer review of "Bicarbonate-Independent Sodium Conductance of Na/HCO3 Cotransporter NBCn1 Decreases NMDA Receptor Function"

_cimb, doi:10.3390/cimb44030086_

Round 1

Reviewer 1 Report

The article submitted by Choi et al., provides data from electrophysiological study on Xenopus oocytes transfected with sodium bicarbonate cotransporter NBCn1, pore-forming NMDA receptor subunits and PSD95. The NBCn1 decreased the glutamate induced currents and prevented from the PSD95-induced augmentation of those currents. The effect was independent from the bicarbonate presence in the buffer.

What is the percentage of the contribution of the sodium current in comparison to the sodium/bicarbonate transport mediated by the NBCn1? In what physiological conditions does it appear?

Further terms of physiological relevance of the study:

Authors propose that increasing the NBCn1 activity might have neuroprotective effects. What would be a strategy, e.g. are there any pharmacological “inducers” for NBCn1?
Additionally, the ref. 14 shows that the opposite approach, namely deletion of NBCn1, has neuroprotective, anti-excitatory effects in KO mice. This discrepancy (stemming from the opposite sodium current effect of NMDA or other mechanism?), should be discussed in more detail in the third paragraph of Discussion.

Author Response

The article submitted by Choi et al., provides data from electrophysiological study on Xenopus oocytes transfected with sodium bicarbonate cotransporter NBCn1, pore-forming NMDA receptor subunits and PSD95. The NBCn1 decreased the glutamate induced currents and prevented from the PSD95-induced augmentation of those currents. The effect was independent from the bicarbonate presence in the buffer.

What is the percentage of the contribution of the sodium current in comparison to the sodium/bicarbonate transport mediated by the NBCn1? In what physiological conditions does it appear?

Response: We appreciate this comment. The percentage of the sodium current relative to cotransport can be calculated by comparing Na ion flux (JNa) and current (A) under a voltage clamp condition. JNa = (dpH/dt)Vβt, where V is an oocyte volume (1 microL) and βt is total buffering power (ie., intrinsic buffering power (βi) + 2.3x[HCO3]i). We did not determine βi, but the reported value is 18.2 mM at 7.05 < pHi < 7.25 (Cougnon et al, 2002 AJP Cell Physiol). In our pH measurement, dpH/dt was 12.6 x 10-5 pH/sec and [HCO3]i was 7.05 in 5% CO2/25 mM HCO3 solution. This gives JNa of 426.284 x 10-14 mole/sec. At -60 mV, NBCn1 channel activity produces 316.3 nA, which equals to 327.83 x 10-14 mole/sec. Thus, the ratio of Na current/cotransport is 0.769; ie, 76.9%. The channel activity constitutively occurs as long as Na+ is available, but it is reduced when membranes are depolarized and close to the Na+ equilibrium potential.

Further terms of physiological relevance of the study:

Authors propose that increasing the NBCn1 activity might have neuroprotective effects. What would be a strategy, e.g. are there any pharmacological “inducers” for NBCn1?
Additionally, the ref. 14 shows that the opposite approach, namely deletion of NBCn1, has neuroprotective, anti-excitatory effects in KO mice. This discrepancy (stemming from the opposite sodium current effect of NMDA or other mechanism?), should be discussed in more detail in the third paragraph of Discussion.

Response: As described in the Discussion, the stilbene derivative 4,4′-diisothiocyano-2,2′-stilbenedisulfonic acid (DIDS) stimulates NBCn1 channel activity. We note a report that DIDS or it analog SITS provides moderate protection against NMDA toxicity and decreases NMDAR-mediated increase in intracellular Ca2+ in cultured rat cortical neurons (Tauskela et al, 2013; ref 43). As for the opposite effects of NBCn1 channel vs cotransport on NMDARs, we revised the fourth paragraph in the Discussion section to address these effects. NBCn1 channel activity decreases NMDAR function in the absence of PSD95 (Figure 2) and there is no apparent difference in IGlu between GluN1·N2/NBCn1 vs. GluN1·N2/NBCn1/PSD95 (Figure 4). Thus, NBCn1 does not need to cluster with NMDARs to decrease the receptor function. On the other hand, the clustering is critical for maintaining NMDAR expression because NBCn1 KO mice develop a significant downregulation of the NMDAR subunit GluN1 and PSD95 in neurons (ref 14). Neurons from these mice display lower apoptotic cell death associated with excitotoxicity and their membranes are less excitable. The downregulation of GluN1 and PSD95, thus, demonstrates that NBCn1 is critical for NMDAR expression during mouse development. We propose that NBCn1 has dual effects on NMDARs. On one hand, it stabilizes NMDAR expression by constituting a large protein complex with the receptors; on the other hand, it decreases NMDAR function by its Na+ channel activity. By producing the dual effects, the transporter can fine-tune the regulation of NMDARs in the CNS.   

Reviewer 2 Report

The research article describes the potential of NBCn1 channel activity for NMDAR regulation to reduce excitotoxity in neuron cells. The study is sound, systematic and supported by statistic evaluation. I just have two remarks:

  • Line 74 and 287, as control 46 nL of sterile water was injected. What is the volume percentage of the injected volume compared to the Xenopus oocyte? Did this not change the osmotic pressure tremendously?
  • I could not find any citation to the journal “Current issues in molecular biology” in the reference list. This seems a bit odd.

Author Response

  • Line 74 and 287, as control 46 nL of sterile water was injected. What is the volume percentage of the injected volume compared to the Xenopus oocyte? Did this not change the osmotic pressure tremendously?

Response: A Xenopus oocyte volume is 1 uL (Parisis, 2012, Mater Methods 2:151). Adding 46 nL becomes 1046 nL and makes the osmolality from 300 mOsM to 287 mOsM. This change is small and negligible. Oocytes can tolerate up to 50 nL of injection without any adverse effect in the function.  

  • I could not find any citation to the journal “Current issues in molecular biology” in the reference list. This seems a bit odd.

Response: In response to this comment, we added two papers from Current Issues in Molecular Biology (ref 44 and 45).